# The Evolving Landscape of Flowcytometric Minimal Residual Disease Monitoring in B-Cell Precursor Acute Lymphoblastic Leukemia

**DOI:** 10.3390/ijms25094881

**Published:** 2024-04-30

**Authors:** Martijn W. C. Verbeek, Vincent H. J. van der Velden

**Affiliations:** Laboratory for Medical Immunology, Department of Immunology, Erasmus MC, University Medical Center Rotterdam, 3015 GD Rotterdam, The Netherlands

**Keywords:** flow cytometry, minimal residual disease, B-cell precursor acute lymphoblastic leukemia, immunotherapy, next-generation flow cytometry, automated analysis, machine learning

## Abstract

Detection of minimal residual disease (MRD) is a major independent prognostic marker in the clinical management of pediatric and adult B-cell precursor Acute Lymphoblastic Leukemia (BCP-ALL), and risk stratification nowadays heavily relies on MRD diagnostics. MRD can be detected using flow cytometry based on aberrant expression of markers (antigens) during malignant B-cell maturation. Recent advances highlight the significance of novel markers (e.g., CD58, CD81, CD304, CD73, CD66c, and CD123), improving MRD identification. Second and next-generation flow cytometry, such as the EuroFlow consortium’s eight-color protocol, can achieve sensitivities down to 10^−5^ (comparable with the PCR-based method) if sufficient cells are acquired. The introduction of targeted therapies (especially those targeting CD19, such as blinatumomab or CAR-T19) introduces several challenges for flow cytometric MRD analysis, such as the occurrence of CD19-negative relapses. Therefore, innovative flow cytometry panels, including alternative B-cell markers (e.g., CD22 and CD24), have been designed. (Semi-)automated MRD assessment, employing machine learning algorithms and clustering tools, shows promise but does not yet allow robust and sensitive automated analysis of MRD. Future directions involve integrating artificial intelligence, further automation, and exploring multicolor spectral flow cytometry to standardize MRD assessment and enhance diagnostic and prognostic robustness of MRD diagnostics in BCP-ALL.

## 1. MRD in BCP-ALL

B-cell precursor Acute Lymphoblastic Leukemia (BCP-ALL) is a hematologic neoplasm originating from the clonal expansion of precursor B-cells within the bone marrow microenvironment [1]. Within the spectrum of childhood malignancies, BCP-ALL is the most prevalent form, with an annual incidence rate of 35 cases per million among children aged 0 to 14 years [2,3]. The long-term survival of pediatric BCP-ALL patients has increased significantly in the past decades [4,5]. A retrospective study of historic therapy studies at the St. Jude Children’s Research Hospital showed that the 5-year survival increased from 21% in the 1960s to 94% in 2017 [4]. Similar survival was observed for pediatric BCP-ALL patients from the Netherlands Cancer Registry, with a 5-year survival from 80% in the 1990s to 91% in the 2010s and with a current survival of 95% [5]. The increase in overall survival throughout the past decades is primarily caused by improved treatment strategies [4,5,6]. Nevertheless, the multiphase treatment protocol, developed in 1962 [7] and including induction therapy, central nervous system (CNS)-directed therapy, consolidation- and maintenance therapy, is still the basis for many clinical protocols today [8].

The presence of chromosomal abnormalities is historically used as a prognostic marker for BCP-ALL. In the 1970s, it was discovered that hyperdiploid BCP-ALL patients had a longer relapse-free survival compared with BCP-ALL patients with a hypodiploid karyotype [9,10]. In subsequent years, novel (cyto)genetic aberrations, including translocations, have been discovered and used for risk stratification [11,12,13], and over 30 genetically distinct subtypes are identified today [14]. Notably, the t(12;21) translocation, resulting in the *ETV6::RUNX1* fusion gene, represents one of the most prevalent genetic abnormalities observed in pediatric BCP-ALL and is associated with a more favorable prognosis. In contrast, the t(9;22) translocation, leading to the *BCR::ABL1* fusion gene, is linked to a poor clinical outcome and is more frequently identified in adult BCP-ALL patients. Beyond these recurrent translocations associated with unfavorable prognosis, alterations in genes crucial for B-cell development, such as *IKZF1* and *PAX5*, further contribute to adverse clinical outcomes [15,16]. 

Traditionally, therapy response in BCP-ALL was evaluated by microscopy-based techniques [17,18]. In this procedure, levels of blasts were accessed by light microscopy or by fluorescence microscopy using B-cell specific markers (e.g., CD10 & TdT), and morphological/clinical remission was defined as the presence of less than 5% blasts in the bone marrow [18,19]. However, it was observed that 25% of the patients in morphological remission eventually relapsed [19], highlighting the need for more sensitive techniques to evaluate residual disease.

In 1993, Potter et al. defined Minimal Residual Disease (MRD) as: “*disease occurring at a subclinical level and beyond detection by conventional methods of assessment*” [20]. This group was the first to correlate the presence of MRD to clinical outcomes, e.g., future relapses. In a retrospective study of 14 children with BCP-ALL, MRD was assessed at day 28, week 12, and/or week 20 and at the end of treatment by PCR amplification of the immunoglobulin (IG) heavy chain (IGH) complementarity-determining region (CDR)-3 region. Of these 14 children, seven reached complete clinical/morphological remission, and seven children relapsed. Bone marrow samples of all patients were positive on day 28. However, in patients who remained in remission, MRD was undetectable at the end of treatment, whereas MRD levels remained positive in relapsed patients. However, a quantitative correlation between MRD levels and outcomes could not be observed.

In later years, three landmark studies [21,22,23] showed that not only is the presence of MRD usable as a prognostic marker, but also that the specific levels of MRD can be used to predict outcomes. Van Dongen et al. [23] showed in a cohort of 210 pediatric BCP-ALL patients that the degree of MRD, assessed by PCR analysis of rearranged IG and T-cell receptor (TR) genes, after 5 and 12 weeks of treatment, correlated with outcome. Patients who were MRD negative after 12 weeks had a 5-year survival of over 90%, whereas patients with MRD levels > 10^−3^ after 12 weeks had a 5-year survival of approximately 30%, and patients with MRD levels > 10^−2^ did not survive for more than 4 years. Similar relations between MRD levels and survival were observed in the study of Cavé et al. [22]. Coustan-Smith et al. [21] conducted a comprehensive evaluation of MRD levels using flow cytometry. MRD levels were monitored at various time points, including post-induction therapy and during weeks 14, 32, and 56 of continuation therapy, as well as after end-of-treatment. Detectable MRD after induction therapy correlated with the presence of cytogenetic aberrancies associated with poor outcomes (*BCR::ABL1* and *KMT2A*-rearrangements). Additionally, the persistence of detectable MRD at all assessed time points was associated with an increased risk of disease relapse. These findings indicated that MRD is a powerful prognostic marker for therapy response, allowing early recognition of both low-risk and high-risk patients. Therefore, since the early 2000s, MRD assessment has been implemented in multiple clinical trials [24]. The AIEOP-BFM ALL 2000 was the first study to adjust treatment strategies based on MRD levels on day 33 after initiation of therapy and at the end of the first consolidation (day 78) [25]. In addition, in the current ALLTogether1 treatment protocol, for patients 1–45 years of age [8], MRD levels are assessed after induction therapy (day 29) by flow cytometry or PCR-based methods. BCP-ALL patients with undetectable MRD levels and lacking a high-risk genetic profile are assigned to a standard-risk consolidation therapy strategy. Patients with detectable MRD but levels below 5% are designated for an intermediate consolidation therapy strategy, while those with MRD levels surpassing 5% are allocated to an intensified high-risk treatment strategy. For patients in the intermediate or high-risk consolidation therapy groups, additional risk stratification is performed through post-consolidation therapy MRD-level assessments.

### 1.1. Molecular Techniques for MRD Assessment

The development of PCR allowed the evaluation of residual disease at a sub-microscopically level [26]. Initially, PCR was used to identify recurrent fusion gene transcripts (e.g., *BCR::ABL1* and *E2A::PBX1*)) in BCP-ALL [27]. This approach allowed a sensitive MRD assessment down to 10^−5^, as reviewed in [28,29]. However, due to the lack of recurrent molecular targets in the majority of BCP-ALL patients, as detected by the classical PCR approach, this technique was only recently applicable in 25–40% of BCP-ALL cases [29]. The introduction of advanced sequencing techniques such as RNA-Seq and whole exome sequencing (WES) allowed the identification of many novel genetic alterations (such as *IKZF1* and *PAX5*) and thereby has significantly expanded the applicability of MRD assessment using genetic alterations to nearly 80% [30].

In B-cell precursors, the IGH locus undergoes recombination of the variable (V), diversity (D), and joining (J) genes, leading to a variety of unique B-cell receptors [31]. Since B-cell malignancies have clonal IG and/or TR gene rearrangements [32], these clonal IG/TR sequences can be used as molecular markers for the detection of MRD in BCP-ALL [29,33]. MRD assessment evaluating IG/TR rearrangements by Real-Time Quantitative PCR (RQ-PCR) is standardized by the EuroMRD consortium (https://euromrd.org/, accessed on 29 March 2024). This standardized approach allows a reliable MRD assessment in 95% of the BCP-ALL patients with a quantitative range down to 10^−4^ and a sensitivity down to 10^−5^ [34]. However, this approach is labor-intensive and requires extensive knowledge and experience [29].

The increasing accessibility of next-generation sequencing (NGS) has facilitated MRD assessment in hematological malignancies [35,36,37,38,39]. At present, NGS is mainly used for the identification of IG/TR rearrangements at diagnosis, where application for MRD analysis is still hampered by less accurate quantification. The use of NGS for evaluating IG/TR rearrangements has the advantage over conventional RQ-PCR in that the assay is not patient-specific and, therefore, less labor-intensive.

### 1.2. Flow Cytometry-Based MRD Assessment

In parallel with the development of molecular techniques, flow cytometry has been applied for MRD assessment in both pediatric and adult BCP-ALL patients. In this review, we will further focus on the laboratory aspects of MRD assessment, especially the development of flow cytometric MRD assays and the impact of targeted therapies on these methods.

### 1.3. Immunophenotype of B-Cell Precursors

B-cell maturation is a tightly regulated process, and B-cell precursors can be characterized based on the expression of various cluster of differentiation (CD)-markers [40]. Generally, CD19 and CD22 are considered B-cell-specific markers and are often used for the primary gating of B-cells [41,42,43]. CD19 is expressed from the pre-B1 stage onwards, whereas CD22 is expressed already in the pro-B cell [40]. The expression of various markers, such as CD10, CD20, CD34, and CD45, changes during B-cell differentiation in a rather constant manner [44]. In contrast, in a large proportion of BCP-ALL patients, the leukemic cells show aberrant expression of these markers [21,45,46,47,48,49,50,51] (Table 1). CD10, expressed by normal B-cell precursors from the pre-B1 stage onwards, is overexpressed in the majority of pre-B-ALL and common ALL patients [45,52,53]. CD10 is not expressed on normal pro-B-cells and, by definition, is also negative on pro-B-ALL cells, which is often associated with *KMT2A*-rearrangements [52,53]. CD34 is expressed on normal hematopoietic stem cells and pro/pre-B1 B-cell precursors [54]. In more than 60% of the BCP-ALL patients, CD34 is aberrantly expressed on BCP-ALL blasts [44,49], although expression can be heterogeneous in some of the patients [55]. CD45 is increasingly expressed during B-cell maturation, but BCP-ALL cells often show decreased CD45 expression [56].

Besides aberrant expression of B-cell maturation markers, non-B-cell markers can be expressed in BCP-ALL [48,50]. Initially, CD13 and CD33, markers of the myeloid lineage (not expressed on normal B-cells), were found to be expressed in 40% of the BCP-ALL patients [47,50]. The concept of aberrant expression of B-cell markers and non-B-cell markers was defined as the expression of “leukemia-associated immunophenotypes” (LAIPs) [21,50], and the presence of LAIPs was observed in 92% of the BCP-ALL patients [50].

**Table 1 ijms-25-04881-t001:** Relevant CD-markers for flow cytometric MRD assessment.

Marker	Expression on Normal Cells	Expression on BCP-ALL Blasts	Remarks	Reference
CD10	Pre-B1 to immature B-cell precursors Mature neutrophils	Overexpressed in 70% of pre-B-ALL and common B-ALL patients	Negative on pro-B-cells and pro-B-ALL cells	[45,52,53]
CD13	Expressed on myeloid cells Negative on lymphoid cells	Expressed in 30% of BCP-ALL cases		[47,50]
CD19	Pre-B1 to mature B-cells Negative on most other leukocytes	Expressed on >90% of BCP-ALL cases	Expression can be lost after CD19-targeted immunotherapy	[41,42,43,57,58,59]
CD20	Expressed on pre-B2-Large cells Highly expressed on immature and mature B-cells Negative on all other leukocytes	Expressed in 40% of BCP-ALL cases		[45,46,47,51]
CD22	Expressed on B-cells from pro-B-cell stage onwards Negative on most other leukocytes	Expressed on >90% of BCP-ALL cases	Can be used as B-cell maker after CD19-targeted therapy	[46,47,60,61]
CD24	Expressed on B-cell precursors Mature neutrophils	Expressed on >80–90% of BCP-ALL cases	Can be used as B-cell maker after CD19-targeted therapy	[60,61,62]
CD33	Expressed on myeloid cells Negative on lymphoid cells	Expressed in 30% of BCP-ALL cases		[47,50]
CD34	Hematopoetic stem cells and early hematopoetic progenitors Pro-B and Pre-B1-cells	Expressed in 60% of BCP-ALL cases	Expression can be heterogenous in BCP-ALL	[44,49,54,55]
CD45	Expressed on all leukocytes	Underexpressed in 30% of BCP-ALL cases		[45,50]
CD58	Expressed on antigen-presenting cells	Overexpressed on >90% of BCP-ALL cases		[63,64]
CD66c	Expressed on myeloid cells Negative on lymphoid cells	Expressed on 36–81% of BCP-ALL cases	Expression correlated with *BCR::ABL* and hyperdiploid cases Negative in *KMT2A*-rearranged BCP-ALL	[65,66,67,68,69]
CD73	Expressed on B-cells, T-cells and folliciular dendritic cells	Overexpressed in 42–70% of BCP-ALL cases	Expression is higher in common- and pre-B-ALL compared to pro-B-ALL patients	[62,70,71,72]
CD81	Highly expressed on B-cells Negative on erytrocytes and neutrophils	Underexpressed in 82% of BCP-ALL cases		[73]
CD123	Expressed on hematopoetic progenitor cells Expressed on CD34-negative B-cell precursors Expressed on plasmacytoid dendritic cells and basophils	Aberrantly expressed in 80% of CD34-positive BCP-ALL cases	CD123 levels are higher in patients harboring a *BCR::ABL* translocation or with an hyperdiploid karyotype	[74,75,76,77,78,79]
CD304	Highly expressed on plasmacytoid dendritic cells	Overexpressed in 40–59% of BCP-ALL cases	Overexpression is associated with *ETV6::RUNX1* and *BCR::ABL* karyotypes	[70,80,81,82,83]

## 2. First Generation Flow Cytometry MRD Panels

The introduction of 3–6 color flow cytometers allowed the design of well-defined antibody panels for MRD assessment and the appropriate identification of LAIPs [21,45,46,48,49,50,84]. These antibody panels used multiple combinations of B-cell differentiation and non-B-cell differentiation markers to detect MRD with a sensitivity down to 10^−4^. All panels included CD19 as the primary gating marker for B-cells, after which other markers were evaluated. The distinction between normal BCPs and BCP-ALL cells was mainly based on the concept of “empty spaces”, which are defined as locations on two-dimensional dot plots where no normal BCP cells are located [85].

Different multi-center and comparative studies showed reliable MRD assessment using limited numbers of antibody combinations [85,86,87,88]. First, Weir et al. [86] designed and validated a protocol using red blood cell-lysed bone marrow samples and two four-color antibody combinations (CD19/CD45/CD20/CD10 & CD19/CD45/CD9/CD34), allowing MRD assessment with a sensitivity of 0.01% in 99% of the BCP-ALL patients. Similar results were obtained by the protocol designed by the BIOMED-1 consortium [85], which used lysed bone marrow samples and five tree-color stainings. In addition to the protocol of Weir et al., the BIOMED-1 protocol included TdT, CD38, and CD22 as additional markers. Mlcáková & Babusíková [87] evaluated marker expression on BCP-ALL cells using a five-tube antibody combination (CD20/CD10/CD45/CD19, CD22/CD34/CD45/CD19, CD38/CD34/CD45/CD19, CD58/CD10/CD45, and TdT/CD10/CD45), used for the diagnosis of BCP-ALL patients. Furthermore, the UKALL Flow MRD group designed a 4-color antibody panel using CD34/CD19/CD10 as backbone supplemented with eight different maturation and aberrancy markers (CD13, CD20, CD22, CD33, CD38, CD45, CD58, and KORSA), respectively [88]. Although these MRD panels worked quite well, the distinction between normal BCPs and BCP-ALL cells remained difficult, especially in samples with high levels of normal/regenerating BCPs, e.g., after the end of induction therapy or after stem cell transplantation [89]. Therefore, flow cytometric MRD assessment had to be further improved, allowing a better separation of normal and malignant BCP cells.

### 2.1. Introduction of Novel Markers

To further improve the flow cytometric MRD analysis, several novel markers were identified and evaluated for their expression on normal and malignant BCPs (Table 1). CD58, a cell adhesion molecule, was highly expressed in BCP-ALL cells compared with CD34-positive BCPs [90]. Ventroni et al. observed that CD58 is expressed in over 99% of BCP-ALL cases, and 94% of these cases showed overexpression of CD58 compared with normal B-cells [63]. Lee et al. [64] showed that CD58 expression decreases during B-cell differentiation and that the majority of BCP-ALL patients overexpress CD58 compared with BCPs. These results indicated that CD58 can be used for flow cytometric MRD assessment.

In addition to CD58, expression of CD81 was studied in the context of BCP-ALL [73]. CD81 was found to be highly expressed in normal BCPs and aberrantly downregulated in 82% of BCP-ALL patients, especially those with a CD34+ immunophenotype. Furthermore, it was shown that CD81 expression remains stable after chemotherapy treatment [73].

To identify novel markers for flow cytometric MRD assessment, Coustan-Smith et al. [62] evaluated gene expression of lymphoblasts from 270 BCP-ALL patients and CD19+CD10+ BCPs originating from four healthy donors by microarray analysis. Thirty genes were found to be differentially expressed, and the expression of these genes was validated by flow cytometry. This resulted in 22 differentially expressed markers (CD44, BCL2, HSPB1, CD73, CD24, CD123, CD72, CD86, CD200, CD79b, CD164, CD304, CD97, CD102, CD99, CD300a, CD130, PBX1, CTNNA1, ITGB7, CD69, and CD49f). Whereas CD72, CD79b, and CD102 were downregulated, all other identified makers were upregulated in BCP-ALL compared with normal BCPs [62]. Remarkably, CD24 and CD44 were upregulated in some patients but downregulated in other patients.

Complementary to this study, Mirkowska et al. [91] evaluated the surfaceome of 19 BCP-ALL cases in a xenograft model via Cell Surface Capture technology. This analysis identified 713 surface proteins, including 181 CD-markers. These CD-markers were compared with gene expression data of sorted normal hematopoietic cells, identifying nine (CD18, CD63, CD31, CD97, CD102, CD157, CD217, CD305, and CD317) potential leukemia-associated markers. Flowcytometric validation highlighted CD97, CD157, CD63, and CD305 as the most informative for the distinction between normal B-cells and BCP-ALL. These studies highlight various potential markers for MRD assessment, and several were subsequently evaluated in other studies (see below).

CD304, a co-receptor for tyrosine kinase receptor signaling, is highly expressed in plasmacytoid dendritic cells [80]. Furthermore, CD304 is overexpressed in 40–59% of the BCP-ALL patients compared with normal BCPs [70,80,81,82,83]. Overexpression of CD304 was stable after chemotherapy treatment [70,80], and overexpression was associated with *ETV6::RUNX1* [81,83] and *BCR::ABL1* [80] translocations. Expression of CD73, a surface enzyme expressed on multiple cell types, was evaluated in BCP-ALL patients; 42–70% of BCP-ALL patients showed overexpression of CD73 compared with BCPs [62,70,71,72]. Furthermore, Wang et al. [71] showed that CD73 expression increased during B-cell maturation, whereas expression of CD73 was higher in common- and pre-B-ALL compared with pro-B-ALL patients, and the expression was stable during treatment [70,71]. CD66c is highly expressed on myeloid cells but not lymphocytes [65]. Flow cytometric analysis showed that CD66c is expressed in part of BCP-ALL patients [65,66,67,68,69]. Tang et al. [69] observed CD66c expression in 81% of BCP-ALL patients, whereas other studies reported frequencies between 36% and 46%. This difference could be explained by differences in patient cohort since Tang et al. only included adult BCP-ALL patients. Expression of CD66c is correlated with *BCR::ABL1* and hyperdiploid cytogenetics [67,68,69], and expression of CD66c was not observed in CD10-negative/*KMT2A*-rearranged BCP-ALL patients [65,68]. The cohort of Tang et al. consisted of 61% out of *BCR::ABL1*-positive patients, explaining the high number of CD66c-positive BCP-ALL patients.

CD123 is an important marker for the proliferation and differentiation of hematopoietic progenitor cells [74]. CD123 is expressed on normal CD34-negative BCPs, but CD123 is not expressed on CD34-positive BCPs [75,76]. However, aberrant expression of CD123 on BCP-ALL blasts was reported, with 80% of the BCP-ALL patients expressing both CD34 and CD123; neither of the antigens was expressed in 11% of patients [75]. Bras et al. [77] evaluated the expression of CD123 in a cohort of 262 BCP-ALL patients, showing CD123-positive blasts in 85% of the BCP-ALL patients [77]. Similar frequencies were found in other studies [74,75], making CD123 an interesting marker for MRD assessment. Furthermore, CD123 levels were higher in patients harboring a *BCR::ABL1* translocation or with a hyperdiploid karyotype [77,78]. Notably, CD123 levels can also be used as a prognostic marker in BCP-ALL since low expression of CD123 is associated with lower overall survival [74,79].

### 2.2. Second Generation Flow Cytometry MRD Panels

The identification of novel markers and the introduction of 6–8 color flow cytometers led to the development of improved protocols for MRD assessment. Denys et al. [92] reported the additional value of a two-tube six-color protocol (CD58/CD19/CD45/CD10/CD22/CD34 and TdT/CD19/CD45/CD10/CD38/CD20) compared with a two-tube four-color protocol (CD34/CD19/CD45/CD22 and TdT/CD19/CD20/CD10). Bone marrow aspirates were acquired at day 15, day 33, and day 78 after the start of therapy. Mononuclear cells (MNCs) were obtained through Ficoll–Isopaque isolation and stained with either the four-color or six-color protocol. Comparison with molecular MRD data showed a concordance of 88% with the four-color tubes and a concordance of 96% with the six-color tubes, showing the added value of the additional markers. The six-color panel allowed an MRD assessment down to 0.002% when measuring 1 × 10^6^ cells/tube. Slightly lower sensitivities (0.006%) were observed by Shaver et al. [93] using a single eight-color tube (CD9/CD10/CD19/CD20/CD34/CD38/CD45/

CD58) and by measuring 4 × 10^5^ MNCs. They proposed a gating strategy in which the first B-cells were selected using a CD19/log side scatter (SSC) gate. Second, plasma cells were excluded using a CD38/log SSC gate. Finally, BCP-ALL cells were identified using the other markers in the panel, and MRD was defined as the proportion of blasts compared with the number of leukocytes. In contrast to this protocol, Borowitz et al. [94] implemented a red cell lysis-based sample preparation. Prepared samples were stained with a two-tube six-color protocol (CD20/CD10/CD38/CD58/CD19/CD45 and CD9/CD13+33/CD34/CD10/CD19/CD45) and a third tube with SYTO-16 for identification of nucleated cells. MRD levels were calculated based on the proportion of BCP-ALL cells compared with the total number of nucleated cells, which allowed MRD assessment to be down to 10^−4^ when measuring 750,000 MNCs. Higher sensitivity was reached by the protocol of Weng et al. [95], which included an eight-color MRD panel containing seven backbone markers (CD58/CD38/CD34/CD10/CD20/CD19/CD45) supplemented with marker(s) in the Phycoerytin (PE)-channel. For these makers, different combinations of CD66c, CD13, CD33, and CD15 were evaluated for MRD assessment of adult BCP-ALL patients. A dilution experiment showed that this panel is able to detect MRD with a sensitivity of 10^−5^ when 2 × 10^6^ MNCs are measured. The same markers were proposed by Bouriche et al. [96], who developed an eight-color panel with a dried antibody mixture, allowing MRD assessment with a sensitivity of 10^−4^ when 200,000 bulk lysed cells were measured. Overall, these panels allow reliable MRD assessment with a sensitivity down to 10^−4^ when sufficient (mononuclear) cells are measured (ranging from 4 × 10^5^ to 1 × 10^6^ cells).

### 2.3. Next-Generation Flow Cytometry Protocols

Previously, flow cytometric MRD levels lower than 10^−4^ were generally defined as MRD-negative. However, Stow et al. [97] evaluated the added value of applying a positivity threshold of 10^−5^ instead of 10^−4^. This retrospective study of 455 pediatric BCP-ALL patients showed that BCP-ALL patients with MRD levels, evaluated by RQ-PCR of IG/TR gene rearrangements, between 10^−5^ and 10^−4^ after induction therapy had a higher risk of relapse than patients with levels lower than 10^−5^ or undetectable MRD (12.7% vs. 5.0%). The EuroFlow consortium aimed to fully standardize flow cytometric MRD assessment for BCP-ALL patients with a sensitivity of at least 10^−5^, similar to PCR-based methods. Therefore, a standardized protocol (sample processing and staining) and antibody panel (two eight-color tubes) were developed [98]. The use of two tubes mitigates the risk of false-positive results attributable to contamination or spill-over effects, as the confirmation of results across both tubes enhances the reliability and accuracy of the MRD assessment. The EuroFlow panel included CD19, CD45, CD34, CD10, and CD20 as backbone markers for the identification of BCPs. CD9, CD123, CD66c, CD81, CD24, TdT, CD58, and surface membrane immunoglobulin light chain kappa/lambda (SmIgK/L) were evaluated for their discriminatory power between BCPs and BCP-ALL cells based on the Automatic Population Separator (APS), included in the Infinicyt software (Cytognos SL; version 2.0.6.b.001). Based on this analysis, CD58, TdT, SmIgK/L, CD22, and CD24 were excluded for use in the BCP-ALL MRD panel [98]. Since both CD66c and CD123 are not expressed by normal BCPs but aberrantly expressed in a subset of BCP-ALL patients (CD66c-positivity: 36–81% [65,66,67,68,69], CD123-positivity: 85% [77]), these markers were combined in the PE-channel. To allow appropriate separation between normal BCPs and BCP-ALL cells in virtually all patients, a second tube with CD73 and CD304 was included. These markers were shown to have a higher discriminating power compared with CD44, CD27, CD164, CD49f, CD200, and CD86. These optimizations resulted in the development of two tubes using CD20-PB, CD45-PO, CD81-FITC, CD34-PerCP-Cy5.5, CD19-PE-Cy7, CD10-APC, and CD38-APC-H7 as backbone and CD66c/CD123 or CD73/CD304 in the PE channel (Table 2). These tubes were used together with a standardized bulk lysis protocol and standardized instrument settings, allowing MRD assessment of BCP-ALL patients with a sensitivity down to 10^−5^ when measuring 4 × 10^6^ cells per tube, with an unprecedented concordance of 98% compared with RQ-PCR. However, this protocol was designed and evaluated solely on chemotherapy-treated patients.

### 2.4. Immunotherapy and Escape in BCP-ALL

In the last decades, immunotherapies have been developed for the treatment of hematological malignancies. In 2017, blinatumomab, a CD3×CD19 bispecific T-cell engager (BiTe), was FDA-approved for the treatment of Philadelphia chromosome-negative relapsed or refractory BCP-ALL in adults and children [99]. Several clinical trials showed longer relapse-free survival (RFS) (7.3 months vs. 4.6 months), longer overall survival (7.7 months vs. 4.0 months), and an improved complete remission rate (34% vs. 16%) within 12 weeks of treatment initiation for blinatumomab treated patients compared with chemotherapy-treated patients [100]. In addition to blinatumomab, CD19-targeted Chimeric Antigen Receptor T-Cell (CAR-T) therapies have been developed to treat BCP-ALL, leading to an improved OS and prolonged RFS [101,102,103,104,105].

Despite these clinical improvements, 40–60% of BCP-ALL patients ultimately relapse after CD19-targeted therapy [57,106,107]. Treatment failure can partly be mediated by the loss of target antigen expression. After CD19-targeted therapy, 10–30% of the patients show CD19-negative BCP-ALL cells at relapse [57,58,59], and these CD19-negative cells are resistant to CD19-directed CAR-T cell therapies [108]. Different mechanisms behind the loss of CD19 have been described. These include mutations that result in ectopic expression of CD19, hyperglycosylation of CD19, and alternative splicing, resulting in loss of the extracellular domain of CD19 [58]. Furthermore, mutations within the *CD81* gene will result in a CD19-negative phenotype since CD81 plays a crucial role in the membrane trafficking of CD19 [109,110]. Hamieh et al. [111] reported loss of CD19 in BCP-ALL through trogocytosis. In this process, CD19 is transferred from the BCP-ALL cells to the bound CAR-T cell, resulting in a >50% reduction in membrane CD19 levels in BCP-ALL cells and to the resistance of CAR-T cell therapy. Besides treatment resistance through epitope loss of CD19, BCP-ALL cells, mainly *KTM2A*—rearranged leukemia, can escape CD19-targeted therapy via lineage switch towards an AML-like phenotype [112,113,114].

With 10–30% of CD19-negative relapses, there is a clear need for novel approaches in flow cytometric MRD assessment for BCP-ALL that do not rely on CD19 as the primary B-cell marker (Figure 1). Since novel treatment strategies use Blinatumomab or CAR-T cell therapies even as first-line treatment for BCP-ALL [115,116,117], the number of patients with CD19-negative relapses may even further increase.

In addition to CD19-targeted therapies, CD22-directed immunotherapies, such as Inotuzumab ozogamicin, have been developed for the treatment of R/R BCP-ALL [118,119]. For example, CD19-targeted therapies and CD22-directed therapy are associated with antigen loss [119]. Given that most (if not all) immunotherapies lead to antigen loss in a subset of patients, it is imperative to consider this factor when developing novel antibody panels for MRD assessment.

### 2.5. Challenges and Novel Approaches for Flowcytometric MRD Assessment

The EuroFlow consortium evaluated if their 8-color BCP-ALL MRD protocol [98] is suitable for MRD assessment without the use of CD19 as a B-cell-specific marker [120]. This study evaluated a common gating strategy primarily based on the expression of CD10 and CD34. This gating strategy allows MRD assessment with a concordance of 89% compared with molecular MRD levels. However, MRD assessment remained challenging in patients with CD19-negative MRD or with ALL-cells that lack CD10 and CD34 expression.

The introduction of >10 color flow cytometers (either conventional or spectral flow cytometers) facilitated the further development of NGF protocols by enabling the inclusion of additional B-cell markers (Table 2). Cherian et al. [60] were the first to report a protocol using CD22 and CD24 as substitutions for CD19 as a B-cell marker. For the gating of B-cells, a “rough B-cell gate” was proposed, and cells that express CD22 or CD24 in the absence of CD66b were defined as B-cells. This protocol, including an alternative gating strategy, showed a high correlation (R^2^ = 0.9975) with the original tube containing CD19, both in samples with CD19-positive and CD19-negative BCP-ALL cells. Mikhailova et al. [61] selected extra B-cell markers, based on the expression of these markers in a cohort of 519 BCP-ALL patients, and they concluded that CD22, CD24, and cytosolic CD79a could substitute CD19 as B-cell specific markers for the identification of BCPs and BCP-ALL. Based on these findings, an 11-color panel was proposed, using CD22, CD24, cytCD79a, and CD19 as B-cell markers. Common maturation makers were included, and SYTO41 was included as a viability stain [121]. Using a CD19-independent gating strategy, based on the expression of CD22, cytCD79a, and CD24, an experimental limit of detection (LOD) of 0.002% and a lower limit of quantitation (LLOQ) of 0.01% was reached when measuring at least 3 × 10^5^ cells. Furthermore, a comparison of the 11-color panel with fusion gene-based PCR showed high concordance for CD19-positive and CD19-negative MRD samples (90.3% and 88.5%, respectively). Singh et al. [122] reported a comparable 10-color panel using CD19 and CD22 as additional B-cell markers, CD38/CD34/CD10/CD20/CD45 as maturation markers, CD58 and CD66c as aberrancy markers and CD13 and CD33 as lineage switch markers. This panel showed an LOD of 0.004% and LOQ of 0.01% when measuring 1 × 10^6^ cells. The panel was validated by a cohort of 65 BCP-ALL patient samples with a concordance of 80% compared with IGH RQ-PCR. Unfortunately, the cohort did not contain patients with CD19-negative MRD. Chatterjee et al. [123] reported a 15-color panel using CD19, HLA-DR, CD22, and CD24 as B-cell markers, together with common maturation and aberrancy markers in combination with a gating strategy without use of CD19. This protocol allowed MRD assessment with a LOD of 0.0004% and LLOQ of 0.001% in patients with CD19-negative MRD when 5 × 10^6^ cells are analyzed. Comparable LOD and LLOQ were observed while using the 14-color protocol designed by Gao et al. [124], which uses CD19, CD22, CD24, and CD86 as B-cell markers in combination with CD66b. Overall, these NGF panels demonstrate that incorporating B-cell markers such as CD22 and CD24 into existing panels, which traditionally rely on CD19 for the identification of (malignant) B-cells, enables reliable MRD assessment down to 0.001% in CD19-targeted BCP-ALL patients, if sufficient cells are required.

**Table 2 ijms-25-04881-t002:** Overview of next-generation flow cytometry panels.

Panel (Reference)	Tubes/ Colors	B-Cell and Maturation Markers											Aberrancy Makers									Viability Dye	# Cells	Minimal Cluster of (MRD) Cells (LOD)	Sensitivity (LLOQ)	Suitable for Targeted Therapy Treated Patients?	Gating Strategy for Normal BCPs and BCP-ALL Cells Provided
		CD10	CD19	CD20	CD22	CD24	CD34	CD38	CD45	cyCD79a	CD81	HLA-DR	CD13	CD33	CD58	CD66b	CD66c	CD73	CD86	CD123	CD304						
Theunissen et al. [98]	2/8																						4 × 10^6^	10	10^−5^	Yes, with adapted gating strategy	Yes, based on expression of CD10 and CD34 [120]
Cherian et al. [60]	2/7 & 8																						5 × 10^5^	NA	NA	Yes	Yes, based on expression of CD22 and CD24 in absence of CD66b
Mikhailova et al. [121]	1/11																					a	3 × 10^5^	10	10^−4^	Yes	Yes, based on expression of CD22, cyCD79a and CD24
Singh et al. [122]	1/10																						1 × 10^6^	20	10^−4^	Not evaluated	Yes, based on expression of CD19 and/or CD22, CD10 and CD22
Chatterjee et al. [123]	1/15																						5 × 10^6^	20	10^−5^	Yes	Yes, based on expression of CD22, CD24 and CD81. Exclusion of myeloid cells via CD33
Gao et al. [124]	1/14																					b	2 × 10^6^	12	10^−5^	Yes	Yes, based on expression of CD22 and CD24 in absence of CD66b

Legend: green: included in the panel; blue/white: not included in the panel; (a): SYTO41; (b): Ghost Dye Violet 510. # means number of cells according international standards.

**Figure 1 ijms-25-04881-f001:**
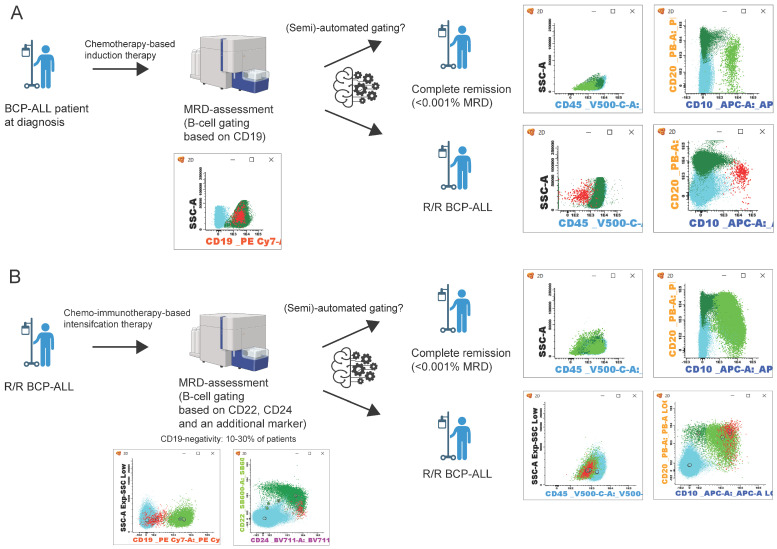
Schematic representation of BCP-ALL patient follow-up. (**A**) follow-up after initial diagnosis and chemotherapy, MRD assessment by B-cell gating based on CD19 expression [98]. (**B**) patient follow-up in relapsed/refractory (R/R) BCP-ALL. After chemo-immunotherapy, B-cell gating relies on the expression of CD22, CD24, and an additional marker [60,121,122,123,124]. Flow cytometry data can be analyzed manually or with a (semi-)automated approach [125,126,127,128,129,130]. After which, patients can be in complete remission (<0.001% MRD) or an R/R BCP-ALL state. Populations in flow cytometry plots: blue: T/NK-cells; dark green: mature B-cells; light green: B-cell precursors; red: BCP-ALL cells. (Figure created with BioRender.com).

### 2.6. (Semi-)Automated MRD Assessment

Next-generation flow cytometry aims to facilitate standardized flow cytometric MRD assessment by standardized sample preparation, instrument setting, and panels. Despite these efforts, data interpretation remains predominantly expert-dependent, necessitating extensive training and experience. To address this challenge and enhance standardization in MRD assessment for BCP-ALL, machine learning-based approaches are being implemented (Table 3; Figure 1). The EuroFlow consortium developed a database-driven, semi-automated, Automated Gating & Identification (AGI)-tool [125]. This AGI tool automatically identifies 15 normal bone marrow cellular subsets in Flow Cytometry Standard (FCS) files of patient samples stained with the EuroFlow 8-color BCP-ALL MRD protocol [98]. Cells not fully compatible with any of the normal cell populations in the database need manual evaluation and may be assigned as MRD [125]. This approach allows reliable MRD assessment in both chemotherapy and targeted therapy-treated patients. Apart from database-driven approaches, Fisher et al. [126] developed an automated tool that uses a combination of hierarchical clustering analysis (HCA) of flow cytometry data in combination with a support vector machine (SVM) to evaluate MRD levels. The HCA algorithm uses Mahalanobis-average linkage to identify clusters within these flow cytometry data. Subsequently, an SVM was trained using flow cytometry data of BCP-ALL patients at diagnosis. The algorithm was validated using a cohort of 123 patients, measured at day 15 after the start of induction, showing a good correlation (R = 0.992) between manual analysis and the HCA/SVM-based algorithm when a cutoff of 0.01% MRD was applied [126]. However, for samples obtained at day 78 after the start of induction therapy, the distinction between BCPs and BCP-ALL cells was less clear due to the presence of normal regenerating BCP cells. An alternative approach for (semi-)automated was developed using the viSNE tool, which uses t-Distributed Stochastic Neighbor Embedding-based methods (tSNE) for analysis of flow cytometry data [127]. For this analysis, data from an FCS file containing viable CD19-positive singlets obtained from the original patient FCS file are clustered based on marker expression. The viSNE tool is able to identify MRD with a sensitivity down to 0.001% when 250,000 B-cells cells are included in the generated FCS file. However, this algorithm requires the pre-selection of B-cells through the expression of CD19. Therefore, this algorithm is not directly suitable for MRD-assessment in CD19-targeted therapy treated BCP-ALL patients. Reiter et al. [128] reported an algorithm that applies multiple Gaussian Mixture Models (GMM) to predict MRD levels at day 15 after start of induction therapy. For this algorithm, a training set consisting of over 300 FCS files of pediatric BCP-ALL patients at day 15 was used. The algorithm is able to automatically identify MRD levels with good concordance to manual MRD levels with a sensitivity down to 0.01%. However, due to the low prevalence of rare BCP-ALL phenotypes (e.g., Pro-B-ALL or lineage switched ALL) in the training set, the predicted MRD levels were underestimated compared with manually obtained MRD levels. More recently, the same group reported an automated tool based on a neural network approach for automated MRD analysis [129]. Similar to the GMM tool, the neural network methodology is trained using FCS files obtained from patients, specifically on day 15 after the start of therapy. Validation of this tool showed a high correlation between the original MRD levels and the predicted MRD levels, with a sensitivity down to 0.01% MRD. However, the GMM and neural network tools were trained and subsequently only applicable for samples obtained on day 15 after the start of treatment (samples with relatively high MRD levels and low numbers of normal (regenerating) BCP cells). A novel software algorithm using radar plots was developed for the automated separation of normal BCP cells and BCP-ALL blasts [130]. In contrast to the GMM and neural network approaches, the radar plot algorithm was trained and optimized using bone marrow samples, which were obtained 29 days after the start of induction therapy, making the tool more suitable for ‘real-life’ MRD assessment.

**Table 3 ijms-25-04881-t003:** Overview of (semi-)automated gating tools for MRD assessment.

(Semi-)Automated Tool	Algorithm	Training Data Set	Input	Sensitivity (LLOQ)	Suitable for Targeted Therapy Treated Patients?	Challenges in Analysis
Verbeek et al. [125]	Database driven Automated Gating & Identification tool	Normal bone marrow samples	FCS-file of MRD sample stained with EuroFlow 8-color BCP-ALL MRD protocol (n = 174)	10^−5^	Yes	Requires manual evaluation of unassigned events (checks)
Fiser et al. [126]	Hierarchical clustering analysis & suport vector machine learning	Leukemic blast populations in diagnostic samples	Raw data from day 15 MRD patient FCS-files (n = 123)	10^−4^	NA	Difficulties with distinction between BCP-ALL cells and normal B-cell precursors
DiGuiseppe et al. [127]	t-Distributed Stochastic Neighbor Embedding-based viSNE		FCS-file containing viable CD19-positive singlets (n = 24)	10^−5^	No, since algoritm requires manual gating of CD19-positive cells	Algorithm requires gating of CD19-postive events prior to automated analysis
Reiter et al. [128]	Gaussian Mixture Models	data from manual gated day 15 patient MRD samples	Data from day 15 MRD patient FCS-files (n = 337)	10^−4^	NA	Only evaluated with MRD samples at day 15
Wodlinger et al. [129]	Neural network approach based on the transformer architecture	data from manual gated day 15 patient MRD samples	Data from day 15 MRD patient FCS-files (n = 519)	10^−4^	NA	Only evaluated with MRD samples at day 15
Shopsowitz et al. [130]	Radar plots	Data from manual gated patient MRD samples (day 29 or later)	Raw data from MRD patient FCS-files (day 29 or later) (n = 20)	10^−4^	NA	

NA = Not assessed.

## 3. Conclusions and Future Perspectives

In the last 60 years, the long-term survival of pediatric BCP-ALL increased from around 20% to over 90% today. This improvement is mainly due to the implementation of treatment strategies and the identification of MRD as a prognostic marker. PCR-based and flow cytometry-based protocols were developed, allowing MRD assessment with a sensitivity down to 1 in 100,000 cells. The development of immunotherapies for the treatment of BCP-ALL challenged the reliability of conventional flow cytometry-based MRD analysis. Technical improvements have driven the development of NGF panels, which include additional B-cell markers to identify BCP-ALL blasts even after the use of targeted therapies. Such NGF approaches can also be used to evaluate the expression of targetable markers on low levels of remaining or recurrent BCP-ALL cells.

In addition to novel protocols, machine learning algorithms have been developed for (semi-)automated analysis of flow cytometric data (Figure 1). However, most approaches are developed for patient samples obtained at an early point after the start of induction therapy (often with relatively high MRD levels and a low number of normal BCP cells), and therefore, these approaches are suboptimal for clinical application. Furthermore, most algorithms have a lower sensitivity (0.01%) compared with manual gating of NGF panels (≤0.001%). However, with the development of new tools, the implementation of artificial intelligence (AI), and the introduction of spectral flow cytometers (>20 colors), automated analysis of flow cytometry data will become increasingly important and feasible in the context of MRD assessment. It can be expected that fully automated algorithms will be developed to analyze flow cytometric MRD data in the near future, which should facilitate the analysis and reduce inter-expert variability, thereby resulting in more robust and reproducible MRD data.

## 4. Take Home Messages

The presence of MRD is the most important prognostic marker in the clinical management of pediatric and adult BCP-ALL.BCP-ALL cells can be distinguished from normal B-cells by abnormal expression of known maturation makers (e.g., CD10, CD20, CD34, and CD45) combined with aberrant expression of other markers (e.g., CD58, CD81, CD304, CD73, CD66c, and CD123).The use of CD19 as a B-cell B-cell-specific marker may become less reliable in the context of CD19-targeted therapies, particularly for patients with loss of CD19.Most next-generation flow cytometry panels include at least CD22 and CD24, along with an additional B-cell marker, for the accurate identification of BCP-ALL cells after CD19-targeted therapies.(Semi-)automated analysis of flow cytometry data likely will facilitate MRD assessment following targeted therapies.

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
