# Peer review of "The Evolving Landscape of Flowcytometric Minimal Residual Disease Monitoring in B-Cell Precursor Acute Lymphoblastic Leukemia"

_ijms, 2024, doi:10.3390/ijms25094881_

Round 1

Reviewer 1 Report

Comments and Suggestions for Authors

I have read the manuscript with great interest; This review critically examines the evolving landscape of flow cytometric minimal residual disease (MRD) monitoring in B-cell precursor acute lymphoblastic leukemia (BCP-ALL), shedding light on recent advancements and their implications for optimizing patient care and treatment outcomes. While this review offers a comprehensive analysis of flow cytometric minimal residual disease (MRD) monitoring in B-cell precursor acute lymphoblastic leukemia (BCP-ALL), there are several minor issues worth addressing to enhance its overall clarity and depth.

To avoid confusion, I didn’t get the purpose or scope of this review includes adult patients, pediatric patients, or both. I think it would be useful to state this from the beginning.

For example in the nomenclature; BCR::ABL1 denotes a fusion of the BCR and ABL1 genes, while BCR::ABL1 designates the corresponding protein product.

Introduction part line 215 What do you mean by “Over 99% of BCP-ALL cases expressed CD58 and 94% of the cases showed overexpression of CD58” please briefly explain.

How often should MRD be measured? How long does MRD monitoring last after the end of treatment? Can a comment be made about this?

Is CD22 antigen loss also observed in patients receiving Inotuzumab ozogamicin, which we can call a targeted agent such as blinatumomab? Does this situation have clinical significance? Or are there problems with MRD measurement after bispecific antibody drug conjugates directed against other epitopes or other surface antigens? What changes should be made in this case? Or what path should be followed?

Similarly, the review should delve into the contributions of minimal residual disease (MRD) assessment following allogeneic hematopoietic stem cell transplantation (alloHSCT), including optimal timing and frequency of evaluations, as well as considerations for utilizing peripheral blood samples versus other sources to maximize diagnostic accuracy and clinical utility.

Author Response

We gratefully acknowledge the Editor-in-chief and the two reviewers for their comments and suggestions that helped us to improve our manuscript.  Below we will indicate point-by-point how we have dealt with these comments. We sincerely hope that these adjustments will make our manuscript suitable for publication in the International Journal of Molecular Sciences

Reviewer: 1

Comments to the Author

I have read the manuscript with great interest; This review critically examines the evolving landscape of flow cytometric minimal residual disease (MRD) monitoring in B-cell precursor acute lymphoblastic leukemia (BCP-ALL), shedding light on recent advancements and their implications for optimizing patient care and treatment outcomes. While this review offers a comprehensive analysis of flow cytometric minimal residual disease (MRD) monitoring in B-cell precursor acute lymphoblastic leukemia (BCP-ALL), there are several minor issues worth addressing to enhance its overall clarity and depth.

Comment:

To avoid confusion, I didn’t get the purpose or scope of this review includes adult patients, pediatric patients, or both. I think it would be useful to state this from the beginning.

Reply:

We thank the reviewer for this comment. MRD assessment is relevant for both pediatric as adult BCP-ALL patients. To highlight this, we added “pediatric and adult” to the abstract (line 13), the age of patients in the ALLTogether protocol (line 103-104), and we added “pediatric and adult” the introduction on flowcytometric MRD assessment (line 140).

Comment:

For example in the nomenclature; BCR::ABL1 denotes a fusion of the BCR and ABL1 genes, while BCR::ABL1 designates the corresponding protein product.

Reply:

We acknowledge this inaccuracy of the nomenclature in the manuscript. All gene symbols are now written in Italics throughout the manuscript.

Comment:

Introduction part line 215 What do you mean by “Over 99% of BCP-ALL cases expressed CD58 and 94% of the cases showed overexpression of CD58” please briefly explain.

Reply:

With this sentence we meant to say that more than 99% of BCP-ALL patients showed an CD58-positive immunophenotype and of these patients, CD58 is overexpressed compared to normal B-cells in 94% of the patients. We acknowledge that the current sentence is unclear and therefore we amended it to: “Ventroni et al. observed that CD58 is expressed in over 99% of BCP-ALL cases, and 94% of these cases showed overexpression of CD58 compared to normal B-cells.”

Comment:

How often should MRD be measured? How long does MRD monitoring last after the end of treatment? Can a comment be made about this?

Reply:

We thank the reviewer for this comment. However, this review focus on the laboratory aspects of flowcytometric MRD assessment rather than the clinical consequences of MRD assessment. Therefore, we respectfully think the addition of these points will be outside the scope of this manuscript. To highlight the scope of our paper, we changed the sentence in line 140-143 to: “In this review we will further focus on the laboratory aspects of MRD assessment, especially the development of flowcytometric MRD assays and the impact of targeted therapies on these methods.”

Comment:

Is CD22 antigen loss also observed in patients receiving Inotuzumab ozogamicin, which we can call a targeted agent such as blinatumomab? Does this situation have clinical significance? Or are there problems with MRD measurement after bispecific antibody drug conjugates directed against other epitopes or other surface antigens? What changes should be made in this case? Or what path should be followed?

Reply:

CD22-targeted therapies are implemented for the treatment of R/R BCP-ALL. Also with use of Inotuzumab ozogamicin, antigen loss is observed. Considering this, the addition of only CD22 will not be sufficient for reliable identification of B-cells. Therefore, multiple B-cells targets need to be included in MRD protocols for targeted therapy-treated BCP-ALL patients. To highlight this, we added the following paragraph: In addition to CD19-targeted therapies, CD22-directed immunotherapies, such as Inotuzumab ozogamicin, have been developed for the treatment of R/R BCP-ALL. Like CD19-targeted therapies, CD22-directed therapy is associated with antigen loss. Given that most (if not all) immunotherapies lead to antigen loss in subset of patients, it is imperative to consider this factor when developing novel antibody panels for MRD assessment.

Comment:

Similarly, the review should delve into the contributions of minimal residual disease (MRD) assessment following allogeneic hematopoietic stem cell transplantation (alloHSCT), including optimal timing and frequency of evaluations, as well as considerations for utilizing peripheral blood samples versus other sources to maximize diagnostic accuracy and clinical utility.

Reply:

We thank the reviewer for this comment. However, as indicated above our review focuses on the laboratory aspects of flowcytometric MRD assessment rather than the clinical consequences of MRD assessment. Therefore, we respectfully think the addition of these will be outside the scope of this manuscript. To highlight the scope of this paper we changed the sentence in line 147 to: “In this review we will further focus on the laboratory aspects of MRD assessment, especially, the development of flowcytometric MRD assays and the impact of targeted therapies on these methods.”

Reviewer 2 Report

Comments and Suggestions for Authors

In this review, the Authors have summarized the state-of-art of flow cytometric (FCM) minimal residual disease (MRD) assessment in acute lymphoblastic leukemia (ALL). There are some formatting issues, especially in references, that must be corrected in further revision.

Introduction section is long, and the Authors start to talk about flow cytometric MRD only on page 4. Moreover, in multiple parts, the Authors do not quickly focus on FCM MRD, while they expose very general concepts on MRD, molecules, or history of MRD.

Take-home messagges are missing, and it would be helpful to add some panels based on international consensus, color staining combinations, or why it should be used a CD marker instead of a different one, and what should guide the choice of CD marker panel making.

There is a lot of information that goes undercovers as it is presented dispersively. Please better focus on FCM MRD.

Tables are small-written and should be provided in better resolution.

Comments on the Quality of English Language

Moderate editing of English language required

Author Response

Manuscript title:       The Evolving Landscape of Flowcytometric Minimal Residual Disease Monitoring in B-cell Precursor Acute Lymphoblastic Leukemia

Manuscript number:  ijms-2964295

Contact information:  Dr. Vincent H.J. van der Velden

Department of Immunology, Erasmus MC

Dr. Molewaterplein 50

3015 GD Rotterdam

The Netherlands

+31 10 704 4253

[email protected]

We gratefully acknowledge the Editor-in-chief and the two reviewers for their comments and suggestions that helped us to improve our manuscript.  Below we will indicate point-by-point how we have dealt with these comments. We sincerely hope that these adjustments will make our manuscript suitable for publication in the International Journal of Molecular Sciences

Reviewer: 2

Comments to the Author:

In this review, the Authors have summarized the state-of-art of flow cytometric (FCM) minimal residual disease (MRD) assessment in acute lymphoblastic leukemia (ALL).

Comment:

There are some formatting issues, especially in references, that must be corrected in further revision.

Reply:

We thank the reviewer for highlighting this formatting issue, we will change the reference style in the revised manuscript, which will solve the formatting issues.

Comment:

Introduction section is long, and the Authors start to talk about flow cytometric MRD only on page 4. Moreover, in multiple parts, the Authors do not quickly focus on FCM MRD, while they expose very general concepts on MRD, molecules, or history of MRD.

Reply:

We appreciate the reviewer's feedback regarding the extensive introduction in our manuscript. While we believe this introduction enhances the reader's understanding of the context, we agree with the reviewer's assessment that certain sections, particularly those covering molecular MRD and the mechanisms of CD19 antigen loss post-targeted therapy, were overly detailed. As such, we have shortened these sections in the revised manuscript.

Comment:

Take-home messagges are missing, and it would be helpful to add some panels based on international consensus, color staining combinations, or why it should be used a CD marker instead of a different one, and what should guide the choice of CD marker panel making.

Reply:

We include the following take-home messages are included at the end of the manuscript:

  1. Take home messages
  • The presence of MRD is the most important prognostic marker in the clinical management of pediatric and adult BCP-ALL.
  • BCP-ALL cells can be distinguished from normal B-cells by abnormal expression of known maturation makers (e.g. CD10, CD20, CD34 and CD45) combined with aberrant expression of other markers (e.g. CD58, CD81, CD304, CD73, CD66c and CD123).
  • The use of CD19 as B-cell specific marker may become less reliable in the context of CD19-targeted therapies, particularly for patients with loss of CD19.
  • Most next generation flow cytometry panels include at least CD22 and CD24, along with an addition B-cell marker, for the accurate identification of BCP-ALL cells after CD19-targeted therapies.
  • (Semi-)automated analysis of flow cytometry data likely will facilitate MRD-assessment following targeted therapies.

Comment:

There is a lot of information that goes undercovers as it is presented dispersively. Please better focus on FCM MRD.

Reply:

As indicated above, we tried to focus on FCM MRD more clearly.

Comment:

Tables are small-written and should be provided in better resolution.

Reply:

We thank the reviewer for highlighting this issue. We submitted Excel-files of the tables which were incorporated in the final PDF-file, which would explain the small tables. For the revised manuscript, we will also upload PDF-files of the table itself, which will probably solve this issue.

Round 2

Reviewer 2 Report

Comments and Suggestions for Authors

Tha Authors have addressed all concerns.